# COST-SENSITIVE HIERARCHICAL CLASSIFICATION THROUGH LAYER-WISE ABSTENTIONS

## ABSTRACT

We study the problem of cost-sensitive hierarchical classification where a label taxonomy has a cost-sensitive loss associated with it, which represents the cost of (wrong) predictions at different levels of the hierarchy. Directly optimizing the cost-sensitive hierarchical loss is hard, due to its non-convexity, especially when the size of the taxonomy is large. In this paper, we propose a **L**ayer-wise **A**bstaining Loss **M**inimization method (LAM), a tractable method that breaks the hierarchical learning problem into layer-by-layer learning-to-abstain sub-problems. We prove that there is a bijective mapping between the original hierarchical cost-sensitive loss and the set of layer-wise abstaining losses under symmetry assumptions. We employ the distributionally robust learning framework to solve the learning-to-abstain problems in each layer. We conduct experiments on large-scale bird dataset as well as on cell classification problems. Our results demonstrate that LAM achieves a lower hierarchical cost-sensitive loss in high accuracy regions, compared to previous methods and their modified versions for a fair comparison, even though they are not directly optimizing this loss. For each layer, we also achieve higher accuracy when the overall accuracy is kept fixed across different methods. Furthermore, we also show the flexibility of LAM by proposing a per-class loss-adjustment heuristic to achieve a performance profile. This can be used for cost design to translate user requirements into optimizable cost functions.

## 1 INTRODUCTION

In many real-world machine learning tasks, the class space has a tree-structured taxonomy, e.g. the organization of species. Biologists have built datasets that include labels ranging from coarse-level domains to the most fine-grained species (Van Horn et al., 2018) . Similar hierarchical structures also exist in the classification of cells (Keren et al., 2018), objects (Yang et al., 2015) and remote sensing images (Li et al., 2017).

When there is a label taxonomy, the ground-truth labels correspond to the leaves of the tree and predicting these labels is known as fine-grained classification. In general, we can predict at any level of the tree, and it is natural to incorporate a cost of such predictions. For example, given an image of "owl", classifying it as either "bird" or "animal" are both correct, but the cost associated with them should be different, since the predicted labels have different levels of specificity. In addition, hard examples are generally easier to distinguish at a coarser level, e.g. in animal mimicry in Fig. 1 (a). In many practical settings, the user is able to specify the cost of different types of errors in a classification problem. The cost can be formulated as an asymmetric matrix with the columns as the labels at the leaves and the rows as the predictions at any level.If we have a label taxonomy, as in Fig. 1 (a), an example of the cost matrix is in Fig. 1 (b).

Cost-sensitive loss is hard to optimize since it is non-smooth and non-convex. Previous works usually rely on its approximation using convex surrogates losses (Zhou & Liu, 2005; Tu & Lin, 2010; Chung et al., 2016; Liu et al., 2019; Khan et al., 2017). However, in hierarchical classification, a convex surrogate for the entire taxonomy is not easy to find. Previous work on hierarchical classification usually takes a top-down strategy to decouple the problem into $O(n)$ binary classification problems for a $n$ node tree (Fan et al., 2015; Charuvaka & Rangwala, 2015; Chen & Warren, 2013). However, there is no guaranteed way to optimize the resulting cost-sensitive loss. The most relevant work that deals with both cost-sensitive and hierarchical learning takes a reward maximization perspective and

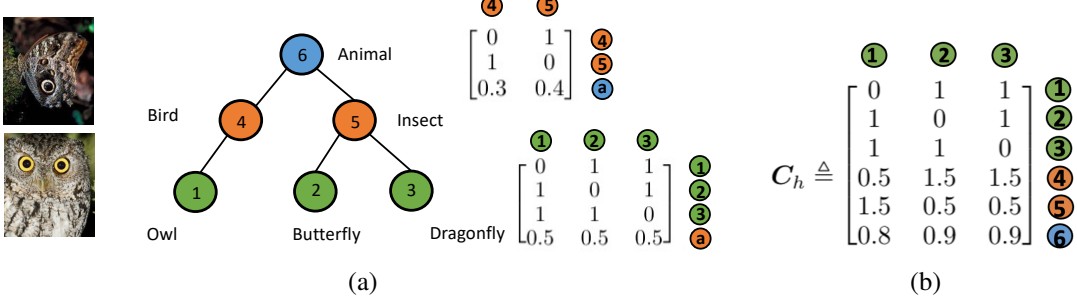

Figure 1: (a) An example of hierarchical species classification on animal mimicry images. The hierarchy is an example of the label taxonomy. The two loss matrices are optimized in the layer-by-layer learning-to-abstain problems in the proposed method. Bold labels are the index of the nodes in the tree. The "a" in the figure represents the abstaining option in each layer in our proposed method. Details can be found in Sec. 3. (b) The corresponding cost-sensitive loss $C_h$ on the full hierarchy.

uses a post-training heuristic to find the desired trade-off between accuracy and specificity, given an overall accuracy (Deng et al., 2012). However, it requires an accurate estimation of probabilities (confidence levels) for the leaf nodes, which is not possible in standard deep neural networks. This is because DNNs tend to be overconfident even when they are wrong (Guo et al., 2017).

Distributionally robust learning framework can be incorporated to optimize a given cost-sensitive loss directly (Asif et al., 2015), which involves solving a linear programming problem at each inference step. However, this scales poorly as the number of classes grows since the time of solving linear programs numerically scales polynomially with the number of classes (Spielman & Teng, 2004). Instead of using a numerical solution, analytical solution to the linear program is available for certain special classes including the abstaining losses when the loss value is no greater than 0.5 (assuming loss is in $[0, 1]$) (Fathony et al., 2016; 2018). Abstaining means that the predictor rejects to predict any of the classes and instead pays a given abstaining cost, which is typically not higher than the cost of wrong prediction. However, these special cases of cost sensitive losses do not incorporate a hierarchy. Moreover, most of the previous methods only discuss the case of low-dimensional linear features and do not incorporate deep neural networks (Asif et al., 2015; Fathony et al., 2016; 2018).

**Our approach:** In this paper, we take a bottom-up approach to break down the hierarchical cost-sensitive loss into the abstaining losses at each layer. Here, abstaining means that the predictor decides not to predict the leaf nodes and instead explores predicting labels at higher levels of the tree. The key insight is that each mistake made by the predictor is due to a series of abstaining and prediction decisions along the hierarchy, which results in a cumulative cost. Fig. 1 (a) demonstrates learning-to-abstain losses in each layer that are equivalent to the hierarchical cost-sensitive loss in Fig. 1 (b). The corresponding prediction mechanism is shown in Fig. 2 (b), where a prediction is first made on the leaves and the abstained data is evaluated on its parents in the tree. We then solve the learning-to-abstain problems in each layer using a distributionally robust learning (DRL) method. Applying DRL to each layer of the tree involves retraining of the neural network for different number of classes. In practice, retraining is efficient since only the heads are replaced and the rest of the representation is shared. Therefore, we train the higher-level models with a different head and parameters from the lower-level ones as the initialization (Fig. 2 (a)).

**Our contribution can be summarized as follows**:

1. We propose to decompose the cost-sensitive hierarchical classification problem into learning-to-abstain sub-problems at each layer. We show the bijective correspondence when the cost-sensitive losses satisfy symmetry conditions (Sec. 3.1).

2. We propose to solve the learning-to-abstain problems using a distributionally robust learning (DRL) approach that directly minimizes the abstaining loss at each of the layers. Our method also allows for efficient training of the model in different layers. We call our method Layer-wise Abstaining Loss Minimization method (LAM).

3. We conduct experiments on large-scale bird recognition with 244 nodes (iNat2017 (Van Horn et al., 2018)) and cell classification tasks (Keren et al., 2018). Our results demonstrate that LAM achieves lower hierarchical cost-sensitive loss in high accuracy regions, compared to previous

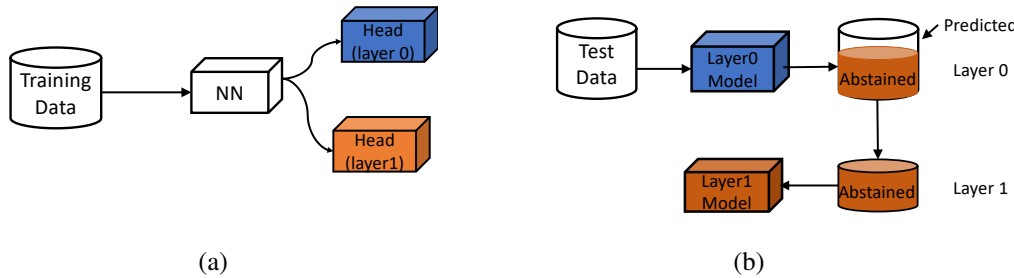

|  |  |
|---|---|
| (a) | (b) |

Figure 2: (a) The training mechanism of the proposed approach. For the training of the non-leaf layers, we relabel the data using the corresponding nodes and initialize the backbone models using parameters from leaf layers, except for the last layer of the neural network. (b) The prediction procedure over the hierarchy. The leaf model is first employed to make predictions on all the samples, then the models on the father layers are employed to make predictions on the abstained samples collected from the previous layers.

methods, even though it is not directly optimizing this loss. This is because the regime of high accuracy corresponds to low abstaining costs and our method is proven to correspond to the original hierarchical loss in this regime. For each layer, we can also achieve higher accuracy when the overall accuracy is kept fixed across different methods.

4. Our method is also flexible enough to achieve a user-defined performance profile efficiently. We provide a heuristic based on per-class loss adjustment to show the potential of our method in achieving arbitrary performance profiles.

## 2 BACKGROUND

**Hierarchical Classification** Given input $x \in \mathcal{X}$ and labels $y \in \mathcal{Y}$, we are interested in learning a classifier $f(x) : x \to y$. We know that $y$ has a taxonomy as a tree structure, which means each $x$ can be classified as a set of correct labels, from fine-grained to coarse labels. Assume the most fine-grained class among all the correct labels for $x$ is $y^*(x)$, then as long as $f(x) \in \boldsymbol{P}(y^*(x))$, the prediction is considered as correct, where $\boldsymbol{P}(y^*(x))$ represents the "path" from the leaf node $y^*(x)$ to the root of the tree. The classifier predicts a coarse label by abstaining from the more fine-grained leaf nodes. However, not all correct labels provide the same correctness. The user defines a cost matrix $\boldsymbol{C}$ that represents the importance of the different types of errors and abstentions when evaluating the performance of $f(x)$. Assume we have $m$ most fine-grained level classes and $n$ total labels in the tree. The matrix has $m$ columns and $n$ rows. The $(i, j)$-th entry $\boldsymbol{C}(i, j)$ in the cost matrix represents the cost the classifier suffers when the ground-truth label of $x$ is $j$ but the prediction is $i$.

**Cost-sensitive Learning** Assume our training data is drawn from a data distribution $P(x, y)$, cost-sensitive learning aims to find a classifier that minimizes the following expected loss function: $f = \arg\min \mathbb{E}_{(x,y) \sim P(x,y)}[\boldsymbol{C}(y, f(x))]$. Under the framework of empirical risk minimization, the expectation is approximated using finite samples. Assume, we have $M$ data points $(x_i, y_i)$ drawn from training set $D_{\text{train}}$, then the problem is converted to: $f = \arg\min \frac{1}{M} \sum_{(x_i, y_i) \sim D_{\text{train}}} \boldsymbol{C}(y_i, f(x_i))$. Conventional machine learning usually approximates the expected cost-sensitive loss using a convex surrogate. Different losses induce different classifiers. For example, the 0-1 loss is a special case of the cost-sensitive loss when there is no label taxonomy. Both log loss and hinge loss are convex surrogates of 0-1 loss. Logistic regression and support vector machines are derived by optimizing log loss and hinge loss, respectively.

**Learning to abstain** Learning-to-abstain problems aim to minimize the abstaining losses using training data. Abstaining loss is a special case of general cost-sensitive losses. It is an asymmetric matrix with $k + 1$ rows and $k$ columns if $k$ is the total number of classes. The additional row

$$\boldsymbol{C}_a \triangleq \begin{bmatrix} 0 & 1 & 1 \\ 1 & 0 & 1 \\ 1 & 1 & 0 \\ \alpha_1 & \alpha_2 & \alpha_3 \end{bmatrix}$$

represents the abstaining cost for each of the ground truth labels. We use $\boldsymbol{C}_a$ to denote an abstaining loss. The abstaining cost should be smaller than the cost of making a wrong prediction. Otherwise, the classifier will never abstain. Then if the first $k \times k$ block of $\boldsymbol{C}_a$ is exactly the 0-1 loss, than the entries $\alpha_k$ needs to satisfy: $0 < \alpha_k < 1$.

## 2.1 DISTRIBUTIONALLY ROBUST COST-SENSITIVE CLASSIFICATION

We now introduce the background in distributionally robust learning (DRL) for optimizing the cost-sensitive loss.

**The formulation:** Fathony et al. (2018) proposed a distributionally robust learning method that incorporate a minimax game to robustly optimize the expected cost-sensitive loss. Assume $g(x)$ and $h(x)$ are valid conditional label distributions. Formally,

$$G : \mathbb{R}^d \mapsto \mathbb{R}^C \triangleq \{g(x) | x \in \mathcal{X}, g(x) \in \mathbb{R}^C \cap \Delta\} \tag{1}$$

Here, $d$, $C$ and $\Delta$ denote the input dimension, class number, and probabilistic simplex, respectively.

Instead of empirical risk minimization, the DRL approach optimizes the following:

$$\arg \min_{G} \max_{H \in \Sigma} \mathbb{E}_{x \sim P(x)}[g^T(x) C h(x)], \tag{2}$$

where $g(x), h(x) \in \mathbb{R}^C$ are the conditional label distributions given an input $x$ and $G$, $H$ are the entire distribution over all the input. $\Sigma$ is a constraint set $h$ needs to also reside in. The idea of this formulation is that $g$, as a predictor player, plays a two-player adversarial risk minimization game Grünwald et al. (2004) with an adversarial player $h$. The predictor player minimizes the expected cost, while the adversary player maximizes the expected cost. The adversary is allowed to perturb the ground-truth labels, subject to certain feature-matching constraints to ensure data compatibility. In particular, the adversary $H$ is constrained by the empirical training data features:

$$\Sigma = \{H | \sum_i h_y \phi(x_i) = \sum_i \mathbb{I}[y_i = y]\phi(x_i), \forall y\}, \tag{3}$$

where $x_i \sim D_{\text{train}}$, $h_y$ is the $y$-th dimension of $h$ and $\phi$ is a feature function. Eq. 3 is a necessary but not sufficient condition for $H$ equals to the ground truth $P(Y|X)$, when $P(Y|X)$ is represented using finite training data. The insights here is that given a predefined feature function $\phi$, when the adversary perturbs the labels, certain aggregate function of $\phi$ on $g$ should equal to the counterpart on the empirical source data.

**The derived prediction form:** Solving Eq. 2 involves solving a two-player zero-sum game with constraints. Fathony et al. (2018) provides analytical solutions for $G$ when the cost-sensitive loss $C$ takes some special cases. In these cases, $G$ would be parameterized with dual parameters $\theta$. Here, we omit the derivation but show an example of the analytical solution when we incorporate $C_a$ in Eq. 2 with $\alpha_k \leq 0.5$.

**Theorem 1.** *(Theorem 13 in (Fathony et al., 2018)) Let $\theta^*$ be the learned parameter, and $f(x)$ be the potential vector for all classes where $f_i(x) = \theta^{*\top}\phi_i(x)$, $i \in [0, C]$. Given a new data point $x$, let $i^* = \arg \max_i f_i(x)$ (break tie arbitrarily), $j^* = \arg \max_{j \neq i^*} f_j(x)$, and $e_{i^*} \in \mathbb{R}^C$ be the $i^*$-th cannonical vector. Then the predictor $G$ can be directly computed as, for each $x$:*

$$g(x) = \begin{bmatrix} e_{i^*} \\ 0 \end{bmatrix} \text{ if } f_{i^*}(x) - f_{j^*}(x) \geq 1 \tag{4}$$

$$g(x) = \begin{bmatrix} (f_{i^*}(x) - f_{j^*}(x))e_{i^*} \\ 1 - f_{i^*}(x) + f_{j^*}(x) \end{bmatrix} \text{ if } f_{i^*}(x) - f_{j^*}(x) < 1. \tag{5}$$

**The parameter learning:** Learning the parameters in DRL involves computing the (sub)-gradients of $\theta$ with respect to the dual objective of Eq. 2. In this paper, we investigate using deep neural networks as the feature function $\phi(x)$ in DRL.

## 3 COST-SENSITIVE HIERARCHICAL CLASSIFICATION

In this section, we describe our proposed distributionally robust learning method for cost-sensitive learning in hierarchical classification. We first break the cost-sensitive loss on the full taxonomy into abstaining losses in the learning-to-abstain sub-problems in each layer (Sec. 3.1). We then propose to use deep DRL to solve the learning-to-abstain problems. We provide details on the gradient (Sec. 3.2). Finally, we propose the algorithms for training and evaluating our cost-sensitive hierarchical classification method (Sec. 3.3).

### 3.1 COST-SENSITIVE LOSS AS AN ACCUMULATIVE LOSS

We take a bottom-up approach and observe that the prediction on the taxonomy takes a series of decisions. In Fig. 1, if an "owl" is classified as a "insect", it means the classifier abstains on the leaf layer and then makes a wrong prediction in the "bird" vs. "insect" layer. Therefore, we can break the overall classification problem into smaller layer-by-layer learning-to-abstain problems. Here we analyze whether there is a one-on-one correspondence between a $C_h$ and a set of $C_a$ in hierarchical classification.

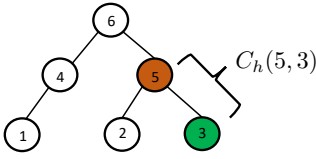

Figure 3: An example of label taxonomy.

Given a tree taxonomy with $m$ leaves and $n$ total number of nodes in the label space, we assume $P(y)$ represents the "path" from the leaf node $y$ to the root of the tree. We also assume $D(y)$ represents the index of the layers of each node. Assume the leaf nodes has $D = 0$. Then we have the following:

**Lemma 1.** *Assume an abstaining loss $C_a$ is associated with each of the layers with $\alpha_i$ as the abstaining loss for each of the nodes, $i \in [1, n-1]$ (except the root node), a hierarchical cost-sensitive loss $C_h$ is uniquely constructed by $C_h(i, j) = \sum_{k \in P(j) \cap k > i} \alpha_k + \mathbb{I}(i \in P(j))$.*

**Lemma 2.** *Given a tree taxonomy, assume the overall cost-sensitive loss is $C_h$. Let $L$ denote the set of leaf nodes in the tree. $D(y)$ is the depth of a node. There is a unique $C_a$ for each layer such that $C_h(i, j) = \sum_{k \in P(j) \cap k > i} \alpha_k + \mathbb{I}(i \in P(j))$, if $C_h$ satisfies the following*

1. *$\forall y \in L$, if $i \in P(y)$, $C_h(i, y) < \min_{k \notin P(y)} C_h(k, y)$;*

2. *$\forall i \notin L$, $C_h(i, j) \leq 1 + D(i)$;*

3. *$\forall i \notin L$, $C_h(i, j) = C_h(i, j')$, for $\forall j, j'$ satisfying $i \in P(j) \cap P(j')$;*

4. *$\forall i \notin L$, $C_h(i, j) = C_h(i, j')$, for $\forall j, j'$ satisfying $i \notin P(j) \cup P(j')$.*

**Remarks:** Lemma 1 is trivial by construction. Lemma 2 can be proved easily by induction. Constraints in lemma 2 defines the structure the cost-sensitive loss needs to possess. Without loss of generality, this lemma shows that our method is applicable to a large set of cost-sensitive losses. Here we explain each of the conditions as follows:

1. The cost of a mistake in any layer is higher than a correct prediction. This is naturally required in any meaningful cost-sensitive losses.

2. The abstaining cost is not greater than the cost of making a mistake in any layer. This makes sure that there is no harm to abstain, compared to predicting a wrong label. Otherwise, no abstention will be made and all predictions and mistakes will be made in the leaf nodes.

3. If two leaf nodes share the same relative position in the tree, i.e., they are the children of the same father, there is symmetry in their cost if they are abstained on. For example, in Fig. 3, if a predictor makes abstention decisions on the leaf layer but predicts node 5, it suffers the same cost when the ground truth is node 2 or 3: $C_h(5, 3) = C_h(5, 2)$.

4. The symmetry also holds if a mistake is made in the non-leaf layers. Following the same logic, in Fig. 3, if a predictor makes abstention decisions on the leaf layer but predicts node 4, it suffers the same cost when the ground truth is node 2 or 3: $C_h(4, 3) = C_h(4, 2)$.

### 3.2 DISTRIBUTIONALLY ROBUST LEARNING FOR LEARNING-TO-ABSTAIN PROBLEMS

We present the distributionally robust learning method for the layer-by-layer learning-to-abstain problems. We first review the (sub-)gradient of the $\theta$ in the original DRL formulation. We then discuss how to incorporate deep neural networks and how parameters in the neural networks are optimized. Finally, we present an algorithm for optimizing the abstaining loss in each layer in Alg. 1.
**Parameter learning in DRL:** According to Fathony et al. (2018), the subdifferential of the objective with respect to the parameter $\theta$ can be fully characterized by the following. For each data sample:

$$s_{\theta_j}(x, y) = \mathbf{conv}\left\{ h_j^* \phi(x) - \mathbb{I}(y = j)\phi(x) \mid h(x) \in H^* \right\}, \tag{6}$$

where $H^*$ is the set of optimal solution of the following:

$$h^* = \arg\max_{h \in \Delta} \min_{g \in \Delta} \left\{ g^\mathsf{T} C_a h + \theta^\mathsf{T} \left[ \sum_j h_j \phi(x) - \sum_j \mathbb{I}(y = j)\phi(x) \right] \right\}. \tag{7}$$

In general, we need solve a linear program to compute $\boldsymbol{h}^*$. When $\alpha \leq 0.5$ in $\boldsymbol{C}_a$, there exists an analytical form of the solution such that the gradient can be computed efficiently.

**Incorporation of deep neural networks:** Instead of low-dimensional features, we incorporate deep neural networks to learn representation $\boldsymbol{\Phi}(\boldsymbol{x}, \boldsymbol{\theta}, \boldsymbol{w})$ for constraining the adversarial $\boldsymbol{h}$ in the DRL framework. The last layer of the network is a linear layer parameterized by $\boldsymbol{\theta}$, the other layers are parameterized with $\boldsymbol{w}$. Formally, $\boldsymbol{\Phi}(\boldsymbol{x}, \boldsymbol{\theta}, \boldsymbol{w}) = \boldsymbol{\theta}^T \phi(\boldsymbol{x}, w)$. Then after we update $\theta$ using Eq. 7 and Eq. 6, the sub-gradient of $\phi(\boldsymbol{x}, \boldsymbol{w})$ can be computed as follows:

$$\boldsymbol{s}_\phi(\boldsymbol{x}, y) = \mathbf{conv}\Big\{ \boldsymbol{\theta} \cdot \Big( \boldsymbol{h}^*(\boldsymbol{x}) - \mathbf{1}(y) \Big) \mid \boldsymbol{h}(\boldsymbol{x}) \in \boldsymbol{H}^* \Big\}, \tag{8}$$

where $\mathbf{1}(y)$ is the one-hot encoding of $y$. Then we use Eq. 8 to back-prop to compute gradients of $w$ in the networks. We present the algorthim in Alg. 1.

---

**Algorithm 1** Training for deep DRL for optimizing the layer-by-layer abstaining loss

1: **Input**: Training data, abstaining loss for this layer $\boldsymbol{C}_a$, DNN $\boldsymbol{\theta}^T \phi(\boldsymbol{x}, w)$, SGD optimizer learning rate $\gamma$, epoch number $T$.
2: **Initialization**: epoch $\leftarrow 0$
3: **While** epoch $< T$
4:     **For** each data sample $(\boldsymbol{x}, \boldsymbol{y})$
5:         Compute $\boldsymbol{h}^*$ following Eq. 7;
6:         Compute the gradient of $\boldsymbol{\theta}$ following Eq. 6;
7:         Back-propagation to compute gradients for $\boldsymbol{w}$.
8:         $\boldsymbol{\theta} \leftarrow \boldsymbol{\theta} - \gamma \boldsymbol{s}_\theta(\boldsymbol{x}, y)$;
9:         Update $\boldsymbol{w}$ by $\text{SGD}_2(\gamma)$ using derived gradients;
10:     epoch $\leftarrow$ epoch $+1$
11: **Output**: Trained networks $\boldsymbol{\theta}^T \phi(\boldsymbol{x}, w)$.

**Algorithm 2** Training and testing of LAM

1: **Input**: Training data $D_{\text{train}}$, test data $D_{\text{test}}$, cost-sensitive loss $\boldsymbol{C}_h$, label taxonomy with $T$ layers, $T$ DNN: $\boldsymbol{M}_0, ..., \boldsymbol{M}_{T-1}$.
2: Decompose $\boldsymbol{C}_h$ into $T$ layer-by-layer abstaining loss $\boldsymbol{C}_a^0, ..., \boldsymbol{C}_a^{T-1}$, pretrain $\boldsymbol{M}_0$ using $\boldsymbol{C}_a^0$;
3: **For** $t \in [0, T-1]$:
4:     Use pretrained $\boldsymbol{M}_0$ to initialize $\boldsymbol{M}_1, ..., \boldsymbol{M}_{T-1}$;
5:     $\boldsymbol{M}_t = $ Algorithm 1($\boldsymbol{C}_a^t, D_{\text{train}}$);
6:
7: //Making predictions:
8: **For** $t \in [0, T-1]$:
9:     $(D_{\text{abt}}, \hat{Y}_t) = $ Predict $(\boldsymbol{M}_t, D_{\text{test}}, \boldsymbol{C}_a^t)$;
10:     $D_{\text{test}} = D_{\text{abt}}$;
11: **Output**: Trained models $\boldsymbol{M}_0, ..., \boldsymbol{M}_{T-1}$, predictions for all the test data.

---

### 3.3 THE OVERALL ALGORITHM

We now present the training and prediction techniques for the whole taxonomy.

**Training:** We run Alg. 1 for each layer of the taxonomy using the corresponding $\boldsymbol{C}_a$ in that layer. However, since the only difference in the data is the relabeling of the labels using nodes in different layers, we initialize the networks in the father layers using pre-trained networks in the leaf layer to achieve efficient training.

**Making predictions:** After the models for each of the layers are trained, the prediction follows a bottom-up procedure. We first use the model in the leaf layer to make predictions on all the samples, the abstained samples are collected for predictions using the father layer's model. Note that the prediction also requires the loss $\boldsymbol{C}_a$ since it involves solving the game, like in Theorem 1. This process goes on until we reach the root of the tree. Eventually, every data sample is assigned to a node in the taxonomy. We summarize the whole algorithm in Alg. 2.

## 4 EXPERIMENTS

In this section, we evaluate the effectiveness of our LAM algorithm in two real-world tasks: bird classification and cell classification. We also compare one-layer LAM with previous baselines on real-world image classification datasets: CIFAR10 and SVHN.

For the bird data, we use a subset of the Aves supercategory data from iNaturalist (iNat) 2018 Competition. The Aves dataset contains images of bird species in the Aves kingdom with labels on the hierarchical taxonomy of the animal in the image: from kingdom to order, family, genus, and species (ground-truth). Thus, there are five layers in the Aves data tree with the Aves kingdom as the

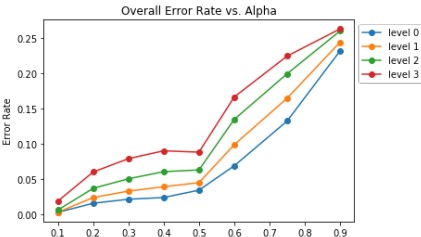 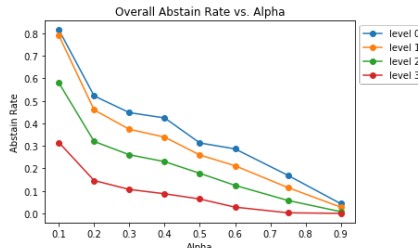

(a) Error rate for different abstaining loss (b) Abstaining rate for different abstaining loss

Figure 4: In the learning-to-abstain problems, the abstaining loss affects the error rate and abstaining rate. The larger the abstaining loss is, the more aggressive the classifier is, thus the larger the error rate and the smaller the abstaining rate.

root and species as leaf nodes. The iNat data was pre-split into training and validation images, where the training data exhibits a long-tailed distribution across classes, and the validation set exhibits a uniform distribution across classes. We include more dataset details in the appendix.

Our cell data comes from the triple-negative breast cancer (TNBC) dataset introduced in Keren et al. (2018). This data comes from multiplexed imaging, a technique that quantifies the abundance of biomarkers within a sample while still capturing the spatial organization of the tissue. Table 1 gives the cell types in the dataset and their respective counts in the training and validation sets. Note that in this data, not all the leaf nodes are in the same level. We then add dummy intermediate nodes to the taxonomy,

Table 1: Distribution of cell types

| Cell Type | Train | Val |
|---|---|---|
| Tumor | 13,264 | 7,489 |
| Immune-Lymphocytes | 2,658 | 672 |
| Immune-Neutrophils | 153 | 318 |
| Immune-Myeloid | 4,115 | 1051 |
| Immune-Other | 1323 | 598 |
| Not-Immune | 2,930 | 967 |

which does not affect the cost structure. More dataset details can be found in the appendix.

CIFAR10 and SVHN are both non-hierarchical or one layer datasets. We use these datasets to validate our advantage on one-layer learning-to-abstain problems over previous baseline Deep Gambler (Liu et al., 2019). To be consistent, we use the same training, validation, and test split as in Liu et al. (2019).

## 4.1 COMPETING METHODS

We use the same abstaining loss for each layer and implement LAM models with $\alpha = [0.1, 0.2, ..., 0.9]$ in Aves classification. In cell classification, we set $\alpha = [0.05, 0.1, 0.15, ..., 0.3]$. We compare our method with DARTS (Deng et al., 2012). DARTS uses the information gain as the reward for each node in the tree. After a model is trained on the leaf nodes, a hyperparameter that adjusts the balance between accuracy and specificity is tuned using binary search. Again, it is not clear what cost-sensitive loss is optimized. In order to make the comparison fair, we implement two versions of DARTS. 1) DARTS (reward): We use the original reward but set the overall accuracy to be the same as every DRL model. 2) DARTS (abstain loss): We modify the reward in DARTS using a "negative cost" so that each node in the tree is associated with a reward that is either $-\alpha$ for nodes in the leaf layer or summation of previous abstaining losses for non-leaf layers. Then we tune the modified version of DARTS using the same overall accuracy achieved by our method. Also, sometimes DARTS cannot achieve the exact required accuracy. We then compare to the closest similar accuracy we can find using DARTS. Note that this correspondence between abstaining losses and rewards is only possible when we use the same $\alpha$ for all the nodes in the same layer. We also compare our method against Deep Gamblers (DG) (Liu et al., 2019). DG minimizes an abstaining loss function based on the doubling rate of gambling–inspired by portfolio theory–in an empirical risk minimization framework. We include more details in the appendix.

## 4.2 RESULTS AND ANALYSIS

**How abstaining loss affects the learning-to-abstain performance?** We plot the overall error rate and abstaining rate of LAM models using different abstaining losses in the Aves classification task's leaf layer in Fig. 4. We see that the larger the abstaining cost is, the more aggressive the classifier is, thus the larger the error rate and the lower the abstaining rate. This validates the working mechanism of the abstaining cost. Imagine the abstaining cost is closer to the cost of making a wrong prediction.

The classifier does not gain much by rejecting to make a prediction. On the other hand, if the abstaining cost is small, the classifier would favor abstentions more.

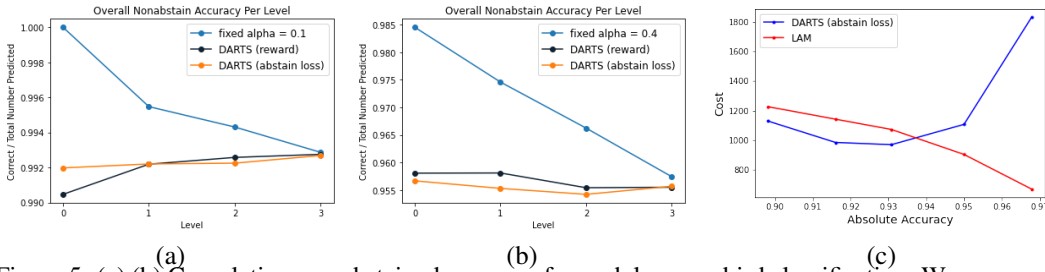

(a)            (b)            (c)

Figure 5: (a) (b) Cumulative non-abstained accuracy for each layer on bird classification. We compare LAM with abstaining cost 0.1, 0.4 with DARTS that achieves a similar overall accuracy; (c) The cost comparison of 5 different LAM models with their corresponding DARTS models achieving the same overall accuracy in the cell classification task.

**How does the overall algorithm perform?** We show the comparison **with DARTS** of non-abstained accuracy on the Aves classification in Fig. 5 (a) and (b), which are the correct samples over all the predicted samples. We can see that, by achieving the same overall accuracy in the whole taxonomy, our method achieves higher non-abstained accuracy consistently in each of the layers. We also compare the cost-sensitive loss achieved by our method and the modified version of DARTS in Fig. 5 (c). We can see that our cost is much lower in the high accuracy domain. We also show the comparison with **DG** of cost-sensitive loss on CIFAR10 and SVHN in Fig. 7 (a) and (b). We see that LAM (with only one-layer) achieves better cost-sensitive loss when the same accuracy is obtained. In Fig 7 (c) and (d), we compare cumulative overall accuracy of equivalent LAM and DG models on Aves classification. The overall accuracy LAM achieves is significantly larger than DG at every level at which a model is trained–when we set DG's loss to be the same as LAM. Note that level 4 denotes the singular root node of the Aves hierarchy, and all inputs abstained on in level 3 are automatically deemed correct in level 4.

**Can our method be used to achieve a specific performance profile?** Since our method can assign different abstaining losses to each class (different $\alpha_k$), we show how we can achieve a similar performance profile by using a loss-adjustment heuristic. Here, the user-defined performance profile means the tuple of overall accuracy, error rate and abstaining rate. In Fig. 5 (d), we use the leaf layer in the Aves taxonomy and start with $\alpha = 0.2$ LAM. Our goal is to achieve a similar performance profile achieved by DARTS with overall accuracy 0.9 in the leaf layer. After four iterations of loss-adjusting, we achieve a very similar performance profile with DARTS.

The loss-adjustment method is a binary search that takes as inputs the error per class profiles of the LAM model (in the first iteration, $\alpha = 0.2$ LAM) and target model (DARTS accuracy 0.9). In every iteration, the LAM model is retrained using an alpha vector that is updated based on a comparison of the error per class between the LAM and target model: if, for a class $k$, the error rate is less than the error rate of the same class achieved by the target model, then $\alpha_k$ for that class is increased to be $(\alpha_{k,\text{current}}+1)/2$. An increase in $\alpha$ corresponds to a lower abstaining and higher error rate as seen in Fig. 4. If the error rate is greater than the error rate achieved by the target model, then $\alpha$ is updated as $(\alpha_{k,\text{current}})/2$. The updated $\alpha$ values are

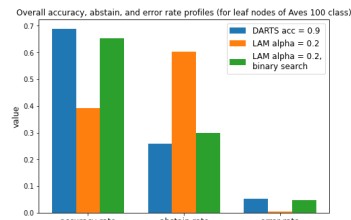

Figure 6: Binary search heuristic to recover DARTS performance.

compiled into an updated vector to retrain the LAM model on. This iterative process stops via a "lookback" condition: at iteration $i$, if the overall accuracy, error, and abstain rates of the previous iteration $i-1$ are all closer in magnitude to the target rates than the current iteration, we stop and use the LAM model trained at iteration $i-1$ as the closest model that achieves the target profile.

## 5 RELATED WORK

**Deep Cost-Sensitive Learning** Traditional cost-sensitive learning works usually focus on transforming data samples (Domingos, 1999), changing the prediction procedure (Langford & Beygelzimer,

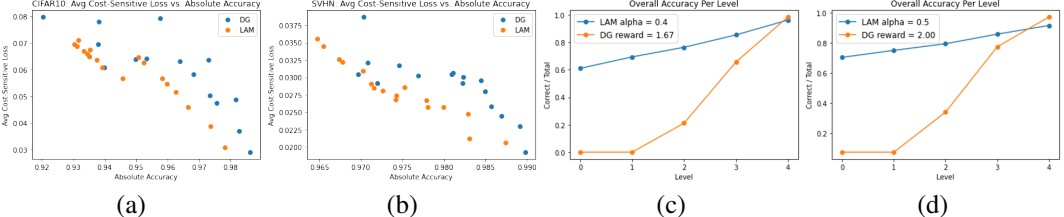

Figure 7: (a) (b) Average cost-sensitive loss vs. corresponding absolute accuracy on CIFAR10 and SVHN. We compare LAM with abstaining costs $0.1, 0.15, 0.2, ..., 0.85$ with DG of equivalent reward. (c)(d) Cumulative overall accuracy for each layer on bird classification. We compare LAM with abstaining cost $0.3$ and $0.4$ with DG of equivalent reward.

2005; Zadrozny et al., 2003), and modifying the existing learning models to include cost-sensitive information (Kukar et al., 1998; Tu & Lin, 2010; Zhou & Liu, 2005; Cao et al., 2013; Gangrade et al., 2021). The investigation of the cost-sensitive perspective in deep learning is still limited. Existing works usually optimize a convex surrogate (Chen & Warren, 2013; Chung et al., 2016; Khan et al., 2017; Charoenphakdee et al., 2021). Learning to abstain can be regarded as a special case of cost-sensitive learning. There are several different strategies. Besides the naive dropout-based method (Gal & Ghahramani, 2016), Geifman & El-Yaniv (2017; 2019) proposed to change the architecture to have a network for deciding abstain or not, while Liu et al. (2019) utilized a tunable weight to incorporate the rejection option in the loss function. In our paper, we investigate cost-sensitive loss on a tree taxonomy and decompose the loss into tractable layer-by-layer abstaining losses, which can be optimized using DRL.

**Deep Hierarchical Classification** For learning with hierarchical classes, most of the work takes the top-down strategy (Babbar et al., 2013; 2016; Zheng & Zhao, 2020; Guo & Zhao, 2021), which suffers from error propagation. Others seek to add an adjustment component (Fan et al., 2015) or hierarchy-related regularization (Gopal & Yang, 2013; Charuvaka & Rangwala, 2015) to the model to make reasonable hierarchical decisions. Deng et al. (2012) is a post-processing method that uses posterior probabilities obtained on a validation set to tune parameters according to information-gain reward and target accuracy. These methods heavily depend on an accurate probabilistic classifier, which is usually not trained under using hierarchical information. Direct improvements over DARTS have included Bayesian methods learning (Jia et al., 2013) and integrating the uncertainty of predicting the parent nodes (Wang et al., 2017), as well as an approach that uses modified top-down and flatten methods to recognize novel classes (Lee et al., 2018). Recently, Zheng & Zhao (2021) proposes to decompose the hierarchical problem in a similar way. However, it stills uses the post-hoc thresholding method for each of the layers.

**Distributionally robust learning** Distributionally robust learning methods have been developed to robustly minimize the general loss functions (Fathony et al., 2016; 2017; 2018). In the most general case, the minimax game is solved using linear programming, which is hard to scale to large number of classes. More advanced techniques have been developed to provide analytical solutions (Fathony et al., 2018) and deal with combinatorially large size of the game (Wang et al., 2015) to optimize the general performance metric. Recently, this framework has been further developed to incorporate deep neural works for optimizing the logarithmic loss under covariate shift (Wang et al., 2021) and make general losses differetiable (Fathony & Kolter, 2020). Our work studies the relation between the general cost-sensitive loss and the learning-to-abstain losses and proposes feasible solutions using the DRL framework to deal with real-world high-dimensional data.

## 6 CONCLUSION

In this paper, we propose to break the overall cost-sensitive learning problem into learning-to-abstain sub-problems in each layer in the tree taxonomy. We provide a distributionally robust learning method for optimizing the abstaining losses in each layer. We show the bijective correspondence between the hierarchical cost-sensitive loss and the set of abstaining losses. We evaluate our method on real-world datasets from applications in bird classification and cell classification. Our method outperforms the competing methods by achieving lower cumulative cost and higher accuracy in each layer. We also provide a heuristic for adjusting abstaining losses to achieve a desired performance profile.

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
