# OpenReview forum: "Cost-Sensitive Hierarchical Classification through Layer-wise Abstentions"
_ICLR.cc/2022/Conference — ICLR 2022 Submitted_

### Official Review · Reviewer_MMjf · 2021-10-31

**Correctness:** 3
**Technical Novelty And Significance:** 2
**Empirical Novelty And Significance:** 3
**Recommendation:** 6
**Confidence:** 5

**Details Of Ethics Concerns:**

I personally deem it has NO such concerns!

**Main Review:**

1.  The strengths:
1.1) using a learning to abstention for CSHC and providing a LAM via efficient optimization of DRL framework layer by layer;
1.2) Comparing DARTS, LAM achieves total boost in performance.
1.3) The strategy is flexible to great extent.
2. The weaknesses:
2.1) Insufficient review on existing works.
There have existed related works devoting to this topic, however, the authors missed them, e.g.,
[m1] Cost-sensitive learning of hierarchical tree classifiers for large-scale image classification and novel category detection.
[m2]Classification with rejection based on cost-sensitive classification.
[m3]Selective Classification via One-Sided Prediction,
While, some of the works the authors have  reviewed are NOT hierarchical classification topic, e.g., (Liu et al., 2019）
2.2) Inappropriate descriptions such as
 For ”most of the previous methods only discuss the case of low-dimensional linear features and do not incorporate deep neural networks“ ，will which new challenges  yield when ”incorporate deep neural networks“ ?
2.3) Lack of the reason why to select DRL for modelling seems NOT to be given! e.g., for its consistency like "Consistent algorithms for multiclass classification with an abstain option"!
2.4) what relation between optimizations at each layer and on all layer is mathematical formulation?
2.5) Insufficient experiments
Why to just use a DARTS (2012) as a baseline, the authors should clarify the reason of just comparing the baseline. In addition, due to the use of DRL in this paper, a strategy different DARTS,  it is unclear whether the boost of performance is from such DRL!

**Summary Of The Paper:**

The submission study the problem of cost-sensitive hierarchical classification (CSHC) with given label taxonomy via learning to abstentions each layer within the hierarchy. Indeed,
1. CSHC subject has had many researches while using abstention or reject decision has also had many works in flat classification, briding both seems relatively few for which the authors present LAM to achieve a so-called new method.
2. Using DRL framework to solve the learning to abstain problems in each layer makes optimization almost decouplable layer by layer, the rationale behind its strategy should attribute to the proved bijective correspondence between the hierarchical cost-sensitive loss and the set of abstaining losses.
3. LAM achieves better performance on limited benchmarks.

**Summary Of The Review:**

In my opinion, incorporating "learning to abstention"  to CSHC is somewhat interesting and new, in particular,  the authors provide the bijective correspondence between the hierarchical cost-sensitive loss and the set of abstaining losses to make the layer-by-layer efficient optimization possible, while the proposed LAM achieves better performance comparing DARTS despite of insufficiency.

---

> ### Author Response · Authors · 2021-11-20
> **Response to Reviewer MMjf**
>
> Thank you for your reviews and feedback. In addition to the general response, we now provide clarifications to your specific concerns:
>
> **Regarding the related work:** We thank the reviewer for the references. We did cite [m1]. We will add [m2][m3] into the related work in our revision. We also discuss the differences between our method and these references here:
>
> [m1] proposes to add an adjustment component that incorporates cost information to regularize the decision-making in hierarchical classification. [m2][m3] only focus on the cost-sensitive learning without any hierarchical structure. [m2] optimizes a surrogate loss in the cost-sensitive learning, while [m3] tries to maximize the coverage with constraints in the error. None of these methods directly optimizes the cost-sensitive matrix. Our proposed method formulates the cost-sensitive hierarchical learning problem as a layer-by-layer abstention problem and directly optimizes the corresponding abstaining losses in each layer. We achieve competitive empirical results in our experiments.
>
> **Regarding “incorporate deep neural networks”:** We want to clarify that incorporating deep neural networks would scale the method up to dealing with high-dimensional data, which makes the method much more practical than the original DRL method. The challenges are in the inference in each optimization step that involves solving the two-player zero-sum game when the potential function is a neural network. We propose to compute the gradients manually using the estimator and the adversary player’s distribution. This enables efficient optimization of a much larger number of parameters in the DRL framework.
>
> **Regarding the choice of DRL:** The reason we use DRL is that it is the only practical method that can directly optimize the abstaining loss/cost-sensitive loss consistently. Even though there are methods in the empirical risk minimization framework that attempt to optimize the surrogate of the cost-sensitive losses, they are not consistent. On the other hand, the method proposed in [a] is not as practical as our deep DRL. We provide additional experimental results to validate this claim. In the figures provided in the general response, we can see that LAM achieves much lower cost-sensitive cost than the baseline. We will add these results to the paper and clarify this in our revision.
>
> [a] Ramaswamy, Harish G., Ambuj Tewari, and Shivani Agarwal. "Consistent algorithms for multiclass classification with an abstain option." Electronic Journal of Statistics 12.1 (2018): 530-554.
>
> **Regarding the relation between optimization in each layer and the overall cost-sensitive loss:** This is a very good question. In our paper, we provide formal bijection mapping between the layer-by-layer abstaining loss and the overall hierarchical cost-sensitive loss. Our next step in the future work includes analyzing whether LAM is consistent with optimizing the overall hierarchical cost-sensitive loss using the DRL framework on the whole tree.
>
> **Regarding additional experimental results:** As mentioned in the general response, we added an additional baseline: using DG to optimize the same abstaining losses in each layer. Note that DG utilizes different “rewards” for prediction and abstaining in an empirical risk minimization framework. We will add a proof to the paper revision of a one-on-one mapping between our abstaining loss $\alpha$ in LAM and the prediction reward in DG. On CIFAR10 and SVHN, we can observe from the results that LAM achieves better cost-sensitive loss compared to DG when the same accuracy is obtained. On Aves, the overall accuracy LAM achieves is significantly larger than DG when we set DG’s loss to be the same as LAM.

---

### Official Review · Reviewer_5LjP · 2021-11-01

**Correctness:** 3
**Technical Novelty And Significance:** 2
**Empirical Novelty And Significance:** 2
**Recommendation:** 5
**Confidence:** 3

**Main Review:**

=========Positives==============
+ It presents a formal approach for the hierarchical classification problems which otherwise somewhat unclear to formalize, have varying performance metrics, and based on ad-hoc methods.


===========Negatives=========
- The significance of the work seems somewhat limited. Even though it is claimed to work on large-scale datasets, this is much smaller than classification problems for hierarchical classification consisting of thousands of labels  [1,2,3]. These datasets also have fat-tailed distribution of instances among labels, and the authors need to check the feasibility of the proposed approach on such settings as well, or discuss the limitations appropriately.
- The main premise of the proposed approach relies on abstention as an option. What if that is not an option, and one needs to make a hard choice as part of the problem formulation.
- The paper relies heavily on existing works such as Distributionally robust cost sensitive classification has been discussed in [4], and the main contribution seems to be extending to hierarchical classification with abstention option. This does not seem enough to be a significant contribution. The formal part of the contribution in Lemma 2 needs to be proven completely instead of leaving it out mentioning simply by use of induction.
- The experimental evaluation is performed against a 2012 approach (DARTS). Is there no work for the last ~10 years which relevant and can be used for comparison. If so, these must be used, otherwise it again seems to go back to the point about the significance of the work and potential impact in the ICLR community.

[1] On Flat versus Hierarchical Classification in Large-Scale Taxonomies, NeurIPS 2013
[2] Recursive regularization for large-scale classification with hierarchical and graphical dependencies, KDD 2013
[3] Learning taxonomy adaptation in large-scale classification, JMLR 2016
[4] Consistent Robust Adversarial Prediction for General Multiclass Classification, (https://arxiv.org/abs/1812.07526)

**Summary Of The Paper:**

The paper presents an approach for hierarchical cost sensitive classification in which abstentions are allowed. It shows a bijection between original cost sensitive problem and the set of layer wise abstaining losses. It is based on using the existing distributionally robust cost sensitive classification and extending to it to the hierarchical setup. The proposed methodology is demonstrated on birds and cell classification datasets, which are claimed to be large-scale. It is compared to a relatively old DARTS method from 2012.

**Summary Of The Review:**

The paper needs to further motivate the significance, evaluate on truly large-scale settings, discuss related works and compare with more recent methods.

---

> ### Author Response · Authors · 2021-11-20
> **Response to Reviewer 5LjP**
>
> Thank you for reviewing and your detailed feedback! We offer some clarifications for your specific comments below.
>
> **Regarding significance and related work:** We thank the reviewer for providing the references. We will add them to the related work section. Here, to emphasize our contribution, we discuss our difference with [1][2][3], respectively:
>
> [1] [3] discuss a tradeoff between top-down strategy vs flat classification but do not consider the cost of different predictions, thus these methods cannot be used in the cost-sensitive setting. [2] proposes incorporating the hierarchical information in the regularization but also cannot deal with cost-sensitive losses. In contrast, our method focuses on optimizing a cost-sensitive loss that is defined on a class taxonomy. We break down the cost matrix into layer-by-layer abstaining losses and use a distributionally robust learning method to robustly optimize them.
>
> **Regarding abstaining as an option:** In the abstaining loss formulation, if we prefer not to abstain, we can just set $\alpha$ = 1. So the general LAM method can accommodate cases when abstaining is not desired beyond a certain level in the tree. However, this may result in inconsistency in the cost-sensitive loss minimization over the whole tree.
>
> **Regarding proof for Lemma 2:** In our paper revision, we will provide an appendix with the proof added.
>
> **Regarding additional experimental results:** As mentioned in the general response, we added an additional baseline: using DG to optimize the same abstaining losses in each layer. Note that DG utilizes different “rewards” for prediction and abstaining in an empirical risk minimization framework. We will add a proof to the paper revision of a one-on-one mapping between our abstaining loss $\alpha$ in LAM and the prediction reward in DG. On CIFAR10 and SVHN, we can observe from the results that LAM achieves better cost-sensitive loss compared to DG when the same accuracy is obtained. On Aves, the overall accuracy LAM achieves is significantly larger than DG when we set DG’s loss to be the same as LAM.

---

> > ### Comment · Reviewer_5LjP · 2021-12-06
> > **post rebuttal**
> >
> > thanks to the authors for the response and adding the additional recent baseline for comparison. However, the problem setup, potential impact and the scale of datasets, overall seems to be of limited impact in the context of ICLR conference.

---

### Official Review · Reviewer_jXQe · 2021-11-02

**Correctness:** 2
**Technical Novelty And Significance:** 1
**Empirical Novelty And Significance:** 1
**Recommendation:** 3
**Confidence:** 4

**Main Review:**

The paper studys cost-sensitive hierarchical classification problems, and aims to propose an efficient method. But the paper contains many issues.
1, The paper claims that "Cost-sensitive loss is hard to optimize since it is non-smooth and non-convex". But how did the paper address this issue? what is the time cost of the proposed methods? The experiments also do not report the time cost the proposed method. Since the paper focuses on the time complexity issues, but the proposed method did not show any benefit of time both from theory and experiments. This is a main major issue.
2, The distributionally robust learning approach is uesed as the optimizer. But the time cost of DRO is also very high. How can you use DRO for fast optimization?what is the time cost？can you report the time in experiments？can the proposed method converge？can you provide both theory and experiments support for convergency？
3, The data sets used in the paper is very small and trivial. Given these data sets, many existing cost-sensitive hierarchical classification methods can be efficiently optimized. Can you test your method on imagenet dataset and larg-scale NLP data sets？
4.There many methods aim to devide the cost-sensitive learning problem into some sub-problems. What is the advantage and disadvantage of the proposed method over existing methods?

**Summary Of The Paper:**

The paper studys cost-sensitive hierarchical classification problems. The novelty is very limited, the experiments are vey tirial. I recommend to reject.

**Summary Of The Review:**

1, The paper claims that "Cost-sensitive loss is hard to optimize since it is non-smooth and non-convex". But how did the paper address this issue? what is the time cost of the proposed methods? The experiments also do not report the time cost the proposed method. Since the paper focuses on the time complexity issues, but the proposed method did not show any benefit of time both from theory and experiments. This is a main major issue.
2, The distributionally robust learning approach is uesed as the optimizer. But the time cost of DRO is also very high. How can you use DRO for fast optimization?what is the time cost？can you report the time in experiments？can the proposed method coverge？can you provide both theory and experiments support for convergency？
3, The data sets used in the paper is very small and trivial. Given these data sets, many existing cost-sensitive hierarchical classification methods can be efficiently optimized. Can you test your method on imagenet dataset and larg-scale NLP data sets？
4.There many methods aim to devide the cost-sensitive learning problem into some sub-problems. What is the advantage and disadvantage of the proposed method over existing methods?

---

### Official Review · Reviewer_9Y3K · 2021-11-04

**Correctness:** 3
**Technical Novelty And Significance:** 2
**Empirical Novelty And Significance:** 3
**Recommendation:** 5
**Confidence:** 3

**Main Review:**

Strengths:
+ The paper has clear motivation.
+ The organization of this paper is good, and it's easy to follow.
+ The proposed method seems to achieve attractive results compared to DARTS.
+ The decomposition into level-wise learning to abstain seems to help find the desired "performance profile" - making it easier by reducing the number of parameters.

Weaknesses:
- I lack comparison with some other methods. Authors claim that _"LAM achieves a lower hierarchical cost-sensitive loss in high accuracy regions, compared to previous methods and their modified versions for a fair comparison"_ but there is really only one baseline method - DARTS. It would be nice to see also the results for some other methods and some simple baseline to get a better idea about the benefits of the proposed approach. As mentioned in the introduction Bayse-optimal solution based on probability estimates from standard DNN would be a nice baseline.
- The proposed prediction procedure over the hierarchy is quite simple and can be applied to any algorithm that can abstain. Comparison with other algorithms within the same prediction procedure would show the benefit of deep DRL.
- The authors use two datasets (Aves, cell classification), but most of the plots present the results only for Aves. The number of datasets is already small. At least, I would like to see all the results for these two.
- Deep DRL is directly based on the work of Anthony et al. 2018.
- Proposed method can be only applied to hierarchies where all the leaves are on the same level.

**Summary Of The Paper:**

The authors propose a new framework for cost-sensitive hierarchical classification. First, they decompose it into level by level learning to abstain (with different abstain costs per class) sub-problems. To solve these subproblems, authors apply deep distributionally robust learning (DRL) approach that directly minimizes the abstaining loss (based on Fathony et al. 2018). These two elements create a method named the Layer-wise Abstaining Loss Minimization method (LAM). The proposed method is compared with DARTS on two datasets and achieves attractive performance. The authors also demonstrate that this decomposition makes it easier to achieve the desired performance profile by adjusting abstaining losses of the layers.

**Summary Of The Review:**

I believe this is a solid work but with a limited scope of contribution. I'm not that familiar with some of the related work and used datasets, but I think additional comparisons are needed to assess the attractiveness of the proposed approach correctly. That is why in my opinion, this work is now marginally below the threshold.

---

> ### Author Response · Authors · 2021-11-20
> **Response to reviewer 9Y3K**
>
> We appreciate your review and feedback. In addition to the general response, here are our responses to your specific concerns:
>
> **Regarding additional experimental results:** As mentioned in the general response, we added an additional baseline: using DG to optimize the same abstaining losses in each layer. Note that DG utilizes different “rewards” for prediction and abstaining in an empirical risk minimization framework. We will add a proof to the paper revision of a one-on-one mapping between our abstaining loss $\alpha$ in LAM and the prediction reward in DG. On CIFAR10 and SVHN, we can observe from the results that LAM achieves better cost-sensitive loss compared to DG when the same accuracy is obtained. On Aves, the overall accuracy LAM achieves is significantly larger than DG when we set DG’s loss to be the same as LAM.
>
> **Regarding “all leaves on the same level”:** LAM can accommodate cases when the leaves are in different levels by adding dummy intermediate nodes. Our cell classification problem in the experiments actually has leaf nodes at different levels. We will add this clarification in our revision.

---

> > ### Comment · Reviewer_9Y3K · 2021-11-30
> > **Re: Response to reviewer 9Y3K**
> >
> > Dear Authors, thank you for your response to my and other reviewers' concerns and for the revision of your manuscript. I compared the revised version with the original submission and checked additional provided materials (plots and appendix). The experiments that compare your approach with another method on new data sets (non-hierarchical, however) and extended related works sections are nice additions. However, I don't find new results especially attractive. LAM does not strictly dominate GD on plots with average cost-sensitive loss vs corresponding absolute accuracy on CIFAR10 and SVHN. And again, as I complained in my review, we don't see all the results (for example, for the overall accuracy per level) for all the methods and datasets on one easy-to-read plot (or in table). I would recommend rewriting the experimental section to present the results in a more coherent way. Finally, I'm concerned by a lack of response to Reviewer jXQe. Because of that, I keep my score as it is.

---

### Author Response · Authors · 2021-11-20
**General Response**

We thank the reviewers for the comments and feedback. Generally, our paper is regarded as “solid”, “has clear motivation”, “interesting and new”, and “easy to follow”. We now address some shared concerns:

**Regarding our contribution:** We want to emphasize that it is non-trivial to solve the cost-sensitive hierarchical classification problem. The previous methods mainly rely on the surrogates of the cost or heuristics of breaking down the problem. Both strategies cannot directly deal with the exact cost information provided by the users. The reason why we can do it is due to the ability of the distributionally robust learning framework to robustly optimize non-smooth/non-convex cost-sensitive losses. But if the cost-sensitive loss matrix is too large in dimension, the inference step is not efficient. We therefore seek to break down the full cost-sensitive loss matrix into layer-by-layer abstaining loss matrices. Our contribution can be summarized as 1) a bijective mapping between hierarchical cost-sensitive loss and the layer-wise abstaining losses; 2) a practical algorithm of distributionally robust learning for the per-layer learning-to-abstain problem and the hierarchical classification problem; 3) competitive empirical results on real-world cost-sensitive hierarchical learning datasets.

**Regarding additional experiments:** As mentioned by reviewers, LAM’s strategy is “flexible to a great extent,” which means we can plug in any method that can abstain. However, we argue that it is not clear whether the cost-sensitive loss is minimized in other methods. We then conduct additional experiments on a recent deep abstaining method: deep gambler (DG) [a]. We demonstrate the results in this anonymous link:
https://docs.google.com/document/d/1Y90Wx-cWGZW_yUJSsmUsZAmFCBgX-OCF3vhi_ceOMDo/edit?usp=sharing

On CIFAR10 and SVHN we aim to compare the performance of different learning-to-abstain methods. We can see that LAM (with only one-layer) achieves better cost-sensitive loss when the same accuracy is obtained. On the hierarchical Aves dataset, the overall accuracy LAM achieves is significantly larger than DG when we set DG’s loss to be the same as LAM. These results will be added to the paper revision.

[a] Liu Z, Wang Z, Liang P P, et al. Deep gamblers: Learning to abstain with portfolio theory[J]. Advances in Neural Information Processing Systems, 2019, 32: 10623-10633.

---

### Author Response · Authors · 2021-11-22
**Summary of changes and paper revision**

We have updated the paper according to the reviews. Here is a summary of the changes:

1. We added the references suggested by the reviewers in the related work and discussed their relations with our work.
2. We added Figure 7 as an additional set of experiments. As mentioned in the general response, we compare with a recent learning-to-abstain method in both one-layer performances on CIFAR10 and SVHN and the overall performance on Aves data.
3. We added more experimental details and proofs in the appendix. Since we got errors when uploading the appendix to the openreivew system, we provide the appendix in an anonymous link here: https://drive.google.com/file/d/1Sm1TaAtOdwu5xQhJnZ6LkkEnQp5-prup/view?usp=sharing.

Please let us know if you have further questions. We are more than willing to answer and resolve them.

---

### Decision · Program_Chairs · 2022-01-20

**Decision:**

Reject

**Comment:**

The authors tackle the problem of cost-sensitive hierarchical classification. They decompose the problem into level-wise learning-to-abstain sub-problems, and apply the distributionally-robust learning (DRL) technique to minimize the abstaining loss. The proposed approach is compared with a few competitors on several data sets. The reviewers find the key idea in the work, namely leveraging DRL as the key technique to solve the decomposed problem, to be somewhat interesting and new. Some of the reviewers find the motivations clear, while others believe that the paper could use a better positioning to connect the motivation with the significance of the technical contributions.

While the authors have extended the discussions on related works and added some additional experiment results during the rebuttal, the reviewers generally agree that the improvements were not sufficient to warrant acceptance. Most importantly, the few baseline competitors and the small-scale data sets make it hard to convince the readers about the validity of the proposed approach. In particular, the scalability of the approach to larger-scale data sets remains questionable, and the spectrum of baseline competitors, both in terms of breadth and recency, is not sufficient. Some reviewers suggest the authors include time/efficiency/convergence analysis of the proposed approach. Furthermore, the authors are encouraged to clarify the significance of contributions, explain the choice of DRL, and deepen the discussions on related works in future revisions.